# Human CD4+ T-Cell Clone Expansion Leads to the Expression of the Cysteine Peptidase Inhibitor Cystatin F

**DOI:** 10.3390/ijms22168408

**Published:** 2021-08-05

**Authors:** Milica Perišić Nanut, Graham Pawelec, Janko Kos

**Affiliations:** 1Department of Biotechnology, Jožef Stefan Institute, Jamova Cesta 39, 1000 Ljubljana, Slovenia; janko.kos@ffa.uni-lj.si; 2Interfaculty Institute for Cell Biology, Department of Immunology, University of Tübingen, Auf der Morgenstelle 15/3.008, 72076 Tübingen, Germany; graham.pawelec@uni-tuebingen.de; 3Health Sciences North Research Institute, 56 Walford Rd, Sudbury, ON P3E 2H2, Canada; 4Faculty of Pharmacy, University of Ljubljana, Aškerčeva Cesta 7, 1000 Ljubljana, Slovenia

**Keywords:** cystatin F, CD4+ T helper cells, cytotoxic lymphocytes, peptidases, granzymes

## Abstract

The existence of CD4+ cytotoxic T cells (CTLs) at relatively high levels under different pathological conditions in vivo suggests their role in protective and/or pathogenic immune functions. CD4+ CTLs utilize the fundamental cytotoxic effector mechanisms also utilized by CD8+ CTLs and natural killer cells. During long-term cultivation, CD4+ T cells were also shown to acquire cytotoxic functions. In this study, CD4+ human T-cell clones derived from activated peripheral blood lymphocytes of healthy young adults were examined for the expression of cytotoxic machinery components. Cystatin F is a protein inhibitor of cysteine cathepsins, synthesized by CD8+ CTLs and natural killer cells. Cystatin F affects the cytotoxic efficacy of these cells by inhibiting the major progranzyme convertases cathepsins C and H as well as cathepsin L, which is involved in perforin activation. Here, we show that human CD4+ T-cell clones express the cysteine cathepsins that are involved in the activation of granzymes and perforin. CD4+ T-cell clones contained both the inactive, dimeric form as well as the active, monomeric form of cystatin F. As in CD8+ CTLs, cysteine cathepsins C and H were the major targets of cystatin F in CD4+ T-cell clones. Furthermore, CD4+ T-cell clones expressed the active forms of perforin and granzymes A and B. The levels of the cystatin F decreased with time in culture concomitantly with an increase in the activities of granzymes A and B. Therefore, our results suggest that cystatin F plays a role in regulating CD4+ T cell cytotoxicity. Since cystatin F can be secreted and taken up by bystander cells, our results suggest that CD4+ CTLs may also be involved in regulating immune responses through cystatin F secretion.

## 1. Introduction

Major histocompatibility complex (MHC) class II-restricted CD4^+^ cells have generally been classified as ‘helper’ T cells (Th), as they help promote or diminish cellular or humoral immune responses by regulating the function of B cells and CD8+ T cells. Nonetheless, data demonstrating that a direct cytolytic role can be ascribed to CD4+ T cells have emerged both in mouse models [1] and humans [2,3]. Furthermore, antigen-experienced CD4^+^ cytotoxic lymphocytes (CTLs) may play an important role in controlling chronic viral infections, such as Epstein-Barr virus (EBV), cytomegalovirus (CMV), and human immunodeficiency virus (HIV) [2,3,4], as well as certain malignancies [5,6]. Recent studies determined that CD4+ CTLs express a specific set of transcription factors that separate them from any known conventional CD4+ Th subset [7,8]. At steady state, only a small number of CD4+ CTLs within effector cells of the intestinal epithelium [7,8] and in the blood of healthy individuals [9] can be detected. However, upon immune challenge, such as viral infections [10], antitumor responses [5,11], and autoimmune disorders [12,13], CD4+ CTLs significantly expand in the blood and peripheral tissues [14]. Under these conditions, their expansion can reach up to 50% of all CD4+ T cells [9], considerably affecting the immune response.

Of the two major cell-killing mechanisms, CD4+ CTLs use the granule-mediated cytotoxic mechanism, rather than the Fas-dependent pathway [15,16,17,18]. Granule-mediated cytotoxicity implies the release of cytotoxic granules containing, most notably, perforin and granzymes upon recognition of the target cells. Granzymes are serine peptidases that trigger cell death upon entering the cell via the pore-forming protein perforin [19]. Accordingly, ex vivo analyses indicate that CD4+ CTLs have lytic granules containing granzymes and perforin and exhibit MHC class II-restricted lytic activity [20,21,22,23,24]. Both perforin and granzymes are synthesized as inactive precursors, which are proteolytically activated by cysteine cathepsins [25,26,27].

Cysteine cathepsins are a group of lysosomal proteolytic enzymes involved in several important aspects of the immune response. An important characteristic in their biology is that their expression and localization patterns in immune cells are tightly linked to their specific function. For example, cathepsin L (Cat L) is involved in the proteolytic processing of extrinsic antigens and self-antigens to antigenic peptides in thymic epithelial cells, whereas cathepsins S and B are involved in antigen processing and presentation in dendritic cells (DCs) [28]. In CTLs, cathepsin C (Cat C) is the major progranzyme convertase that activates progranzymes by removing a dipeptide at their N-terminal end [26]. In the absence of Cat C, cathepsin H (Cat H) acts as an additional progranzyme convertase [27]. Furthermore, Cat L plays a role in activating perforin precursors [29,30] by cleaving their C-terminal ends [25,30]. Recent studies have also implicated Cat L [31] and legumain as important regulators of T helper cell subset differentiation [32].

Apart from regulation through their expression, processing, and localization, the activity of cysteine cathepsins is tightly regulated by their endogenous inhibitors, cystatins. Cystatin F is an inhibitor of cysteine peptidases expressed solely in immune cells [33]. It is the only type-II cystatin capable of entering endosomal/lysosomal vesicles in which it directly regulates the activity of intracellular cysteine cathepsins [34]. The inhibitory profile of cystatin F depends on its molecular form. The expression of cystatin F is regulated by the transcription factor CCAAT/enhancer-binding protein alpha (C/EBPα) [34,35] and depends on the activation state or differentiation status of the cells. After synthesis, cystatin F forms disulfide-linked dimers that do not inhibit the C1 family of cysteine peptidases but inhibit legumain, a peptidase from the C13 peptidase family [36]. N-terminal cleavage within endosomes/lysosomes results in active, monomeric cystatin F that is a strong inhibitor of Cat C and Cat H [37]. Glycosylation is also an important feature of cystatin F, as it directs its uptake from the extracellular space, translocation to endosomes/lysosomes, and subsequent activation [33,38,39].

Cystatin F was shown to attenuate the granzyme-mediated cytotoxicity of natural killer (NK) cells by regulating the intracellular activities of Cat C and Cat H [33]. Furthermore, cystatin F is an upstream regulator of split anergy in NK cells [40], and its increased expression was detected in anergic CD8+ T cells [41]. In human CD8+ T cell blasts, cystatin F co-localizes with granzyme A, perforin, and lysosomal-associated membrane protein-1 [37]. Furthermore, extracellular cystatin F, secreted by bystander cells, attenuates granzyme-mediated cytotoxicity in both NK and CD8+ T cells [33,42]. Here, we describe for the first time cystatin F expression in CD4+ T-cell clones together with the components of cytotoxic granules. In CD4+ T-cell clones, the main targets of cystatin F were the granzyme convertases cathepsins C and H. This suggests that, apart from regulating cytotoxicity in NK cells and CD8+ T cells, cystatin F could represent an additional layer of regulation of CD4+ CTL effector function.

## 2. Results

As previously described [43], long-lived T-cell clones were found to be exclusively CD4+ and to carry αβ3 antigen T-cell receptors (TCR2). They were grouped into three different age periods according to the number of population doublings (PDs; as calculated from cell expansion factors): (1) “young” (<30 PDs), (2) “middle-aged” (30–43 PDs), and (3) “old” (>43 PDs).

### 2.1. Cystatin F Is Expressed in Long-Term Cultured Human CD4+ T-Cell Clones

Using Western blot analysis, we have shown that cystatin F is expressed in all the lymphocyte clones analyzed (Figure 1A, Appendix A). Under non-reducing conditions, both monomeric and dimeric forms of cystatin F were present as multi-band patterns due to heterogeneous N-glycosylation [44] (Figure 1A). The total cystatin F content in the majority of samples from the “middle-aged” (30–43 PDs) group of lymphocyte clones was higher than in samples from either “young” (<30 PDs) and “old” (>43 PDs) groups of lymphocyte clones. Within the “middle-aged” clones, the total cystatin F levels varied markedly between different individuals. The ratio of the active monomeric form of cystatin F compared to total cystatin F was higher in “middle-aged” and old T-cell clones than that in the “young” clones. The expression of C/EBPα was also detected in all of the lymphocyte clones analyzed, and its expression level varied between the clones of different ages as well as between the clones with the same PDs but from different individuals (Figure 1B). As shown before, upon digestion with PNGase F (which cleaves nearly all N-linked oligosaccharide chains), cystatin F migrates as a single form [37]. Digestion with EndoH (which cleaves N-linked glycans between the two *N*-acetylglucosamine residues in the core region of the glycan chain in high-mannose and hybrid, but not complex, glycans) revealed that cystatin F expressed in CD4+ T cells contains both high-mannose and complex glycans. The presence of high-mannose glycans, necessary for lysosomal targeting of cystatin F, is in line with the significant content of monomeric, active cystatin F detected in CD4+ cells. Furthermore, similar glycosylation patterns in both young and old analyzed CD4+ T-cell clones suggest that the age of T-cell clones does not affect cystatin F glycosylation (Figure 1C).

### 2.2. The Main Targets of Cystatin F in Long-Term Cultured CD4+ T-Cell Clones Are Cathepsins C and H

The main progranzyme convertases Cat C and Cat H in CD4+ T-cell clones were expressed in all samples analyzed with Western blot. Cat C is synthesized as a 60 kDa zymogen that undergoes proteolysis at several sites to release an internal propeptide. The resulting mature Cat C monomer is composed of three tightly linked subunits, a 16 kDa N-terminal exclusion domain, a 23 kDa catalytic heavy chain, and a 7.5 kDa light chain, which are held together by non-covalent interactions. The relevant protease(s) responsible for the physiological maturation of pro-Cat C have not yet been identified in humans. However, human recombinant Cat L and Cat S activate purified pro-Cat C in vitro [45]. The expression of pro-Cat C and the 23 kDa catalytic heavy chain was the highest in young CD4+ T-cell clones (Figure 2A). The expression of the mature form of Cat C, except in the oldest T-cell clones, declined with age. The expression of the mature form of Cat H was generally the highest in younger clones but varied significantly between clones of different ages (Figure 2B). The processing of pro-Cat H is a multistep process that is distinct from that of other cathepsins. It proceeds from an inactive 41 kDa pro-form, through a 30 kDa intermediate, to the 28 kDa mature enzyme [46]. It was reported that the level of processing differs significantly between different human cell lines [46] and is possibly very sensitive to changes in T cells with increasing age. Immunoprecipitation with anti-cystatin F antibodies and subsequent detection showed that the main interacting partner of cystatin F in CD4+ T cells are cathepsins C and H (Figure 2C,D, Appendix A).

### 2.3. Human Primary CD4+ T-Cell Clones Express Legumain and Cathepsin L

In all samples of CD4+ T cells, the Cat L pro-form (38 kDa) and single-chain form (SC, 27 kDa) were detected (Figure 3B). Cat L is synthesized with a prodomain that protects against premature activation in the endoplasmic reticulum and Golgi and is targeted to the more acidic endosomes in which it undergoes autolysis to generate active SC-Cat L [47]. Whereas the majority of Cat L was in the SC form in young and old T-cell clones, the pro-Cat L signal was pronounced in most middle-aged clones. Legumain (also known as asparaginyl endopeptidase) is a lysosomal cysteine protease that converts SC-Cat L to disulfide-bonded TC-Cat L (20 kDa); both forms are generally thought to be biochemically active [48]. Legumain is produced and secreted as inactive prolegumain (56 kDa) and processed into the enzymatically active 46 and 36 kDa forms as well as a 17 kDa enzymatically inactive C-terminal fragment [49]. In mice, legumain has been suggested to be responsible for the processing of the SC forms of endosomal Cat L, Cat H, and Cat S [50]. Although significant expression of pro- and mature-form legumain was detected in human CD4+ T cells, the two chain (TC) Cat L form was not detected in any of the samples analyzed (Figure 3A).

### 2.4. Human Primary CD4+ T-Cell Clones Express Active Granzymes B and A

Next, we examined whether CD4+ T cells possess active granzymes. The expression of granzyme B varied between the cell clones of different PDs and was generally more pronounced in the middle-aged T-cell clones (Figure 4A). The activity of granzyme B detected in the whole-cell lysates of CD4+ T cells was the highest in the old T-cell clones. Granzyme A expression was less pronounced and declined with clone age; however, its activity, similar to granzyme B, increased with age (Figure 4B,C). In all CD4+ T-cell clones examined, the expression of another cytotoxic effector molecule, perforin, was detected (Figure 4A, Appendix A). However, its unprocessed pro-form was more pronounced in all CD4+ clones, and its expression varied between clones of different age.

## 3. Discussion

The acquisition of cytotoxic functions by CD4^+^ T cells was first described in T cell lines and clones generated in vitro. However, multiple studies have demonstrated that antigen-specific CD4^+^ T cells possess direct MHC class II-restricted cytotoxic activity in vivo [16,51,52,53,54,55,56]. CD4+ human T-cell clones were derived from activated peripheral blood lymphocytes of healthy young adults to establish cloning efficiencies and clonal longevities [57]. Their longevity and phenotypic characteristics have been previously described [43]. Here, we report that the CD4+ human T-cell clones have different cathepsin signatures from conventional naïve CD4+ T cells, expressing the components of the cytotoxic machinery characteristic for CTLs as well as the cysteine protease inhibitor cystatin F, which has been previously implicated [33,42] in regulating the cytotoxic function of CTLs.

In addition to CTLs, cystatin F expression was detected in eosinophils, neutrophils, mast cells, DCs, and macrophages [44,58,59,60,61,62]. Through regulating cysteine peptidases, critical for normal granule biogenesis, cystatin F controls eosinophil survival [60]. In DCs, cystatin F was proposed to regulate the activity of Cat L, thus controlling the processing of procathepsin X, which promotes cell adhesion [63]. In CD4+ T-cell clones, its expression was detected in all samples analyzed, suggesting that the induction of cystatin F expression starts relatively early upon stimulation. The proportion of the active form was the highest in older T-cell clones, suggesting either an increase in the activity of the activating peptidase of cystatin F or, alternatively, an increase in cystatin F secretion. Although it was previously shown that clone behavior is independent of the origin of the T-cell clones (56), cystatin F expression and the presence of its active form varied between the clones with the same PDs indicating its dependence not only on culture conditions but also on donor variables such as age and/or sex. Further studies are needed to support this observation. We failed to detect differences in glycosylation between young and old T-cell clones that could suggest increased cystatin F sequestration to secretory pathways. Thus, the increased expression of active cystatin F is most likely a consequence of its increased activation. The detection of C/EBPα expression in T-cell clones is in accordance with its role as a regulator of cystatin F expression [35]. C/EBPα is normally expressed in myeloid-lineage cells, however in human CD4+ T cells C/EBPα expression was documented in PD-1+ memory phenotype CD4+ T-cell subpopulations [64] and in mouse follicular Th. High expression of C/EBPα was found to be necessary for restriction of IFN-γ expression in T cells and induction of proper class switching by B cells [65].

Similarly, the expression of Cat C is restricted to CTLs and neutrophils in which it activates granule-associated serine proteases and antigen-presenting cells [45]. Cat H is expressed in CTLs and most antigen-presenting cells [66]. These cathepsins, normally not present in CD4+ T cells, were found to be expressed and processed to their mature forms in all the T-cell clones analyzed in the current study. Data on Cat C and Cat H expression in human CD4+ T cells is missing; however, significantly upregulated Cat C and Cat H genes were found in mouse IL-17-producing T helper (Th17) cells compared with Th0 and T regulatory lymphocytes [32]. In line with previous studies showing that cystatin F colocalizes with Cat C and Cat H in the cytotoxic T cell line Tall-104 [37,41], we have here demonstrated that the main interacting partners of cystatin F are Cat C and Cat H.

Naïve human CD4+ T lymphocytes do not express Cat L [32,67]. However, in mice, in addition to cytokines and transcription factors, the differentiation of CD4+ cells to Th17 cells is actively regulated by Cat L [32]. The *CSTL1* gene is upregulated during the early stage of Th17 cell differentiation [31] and has recently been characterized as a promising marker for human Th17 cell identification [67]. Furthermore, it was shown that Cat L of human but not murine CD4+ T lymphocytes processes complement component 3 (C3) to active C3b and C3a; the latter is required for homeostatic T- lymphocytes survival and optimal production of Th1 cytokines [68]. Besides Cat C and Cat H, active monomeric cystatin F can inhibit Cat L and thus affect the processing of perforin from its precursor form. Despite the high affinity of cystatin F for Cat L, we failed to confirm their interaction in CD4+ lymphocyte lysates. However, cystatin F can affect Cat L activity indirectly, by regulating legumain activity. The second active site on the cystatin F molecule is involved in inhibiting legumain, and the dimeric form of cystatin F is also an active inhibitor of legumain. In addition to other functions, it was described that legumain processes SC-Cat L into its active TC form in human CD4^+^ T lymphocytes [68]. In bone marrow-derived macrophages, the internalization of exogenous cystatin F inhibits legumain and causes the accumulation of SC-Cat L [39]. Inhibiting legumain activity in human CD4+ T lymphocytes reduced the appearance of the Cat L active forms, the generation of C3a in T lymphocytes, and the induction of IFN-γ-secreting cells by approximately 50% [68]. Decreased legumain activity in CD4+ T lymphocytes, due to high cystatin F content, could therefore regulate the activity of Cat L by degrading its SC forms. Considering that the TC form of Cat L was not detected in cell lysates from CD4+ T cells, it is likely that, besides Cat C and Cat H, one of the main targets of cystatin F in CD4+ T cells is legumain. Whereas in *Serpinb1-deficient* mice, legumain inhibition negatively controlled Cat L-mediated Th17 induction without affecting Th1 responses, findings in human CD4+ T lymphocytes suggest that legumain functions upstream of the Cat L-C3 axis and is an integral part of human Th1 initiation [32,68].

Perforin expression has been reported previously in CD4+ T cells [69,70]. In mice, some perforin-expressing effector CD4+ T cells protect against lethal influenza infection [71,72]. Perforin expression has also been demonstrated in human CD4+CD25+ T-regulatory cells [73]. The conditions under which naive CD4+ T cells are initially activated can increase their potential to express perforin [72]. For example, perforin mRNA is undetectable in CD4+ T lymphocytes cultured under Th2-inducing conditions [74], but CD4+ T lymphocytes differentiated under Th1-inducing conditions can express detectable amounts of perforin mRNA [69]. The T-cell clones studied here were most closely aligned with an IFN-producing Th1 phenotype [75].

In Jurkat cells, CD4+ co-receptor expressing lymphocyte cell line stimulation of the NF-kB-signaling pathway is involved in controlling granzyme B expression [76]. However, our results show that stimulating the NF-kB-signaling pathway Jurkat cell does not induce the expression of perforin or cystatin F (Appendix A). The expression of perforin and granzymes in immune cells other that CTLs depends on different signals and is not necessarily concomitant. For example, human myeloid DCs upregulate perforin protein expression upon stimulation with TLR7/8 ligands and release perforin together with granzyme B [77]. However, no such upregulation is found in plasmacytoid DCs (pDCs) stimulated with TLR7/8 ligands or interleukin (IL)-3/-10 [77]. Demethylation of specific regulatory elements was found to induce perforin overexpression in CD4+ T cells from lupus patients [78]. Human B cells also express granzyme B, which was found to be induced by IL-21, though in the context of other stimuli such as viral antigens [77]. Human macrophages, basophils, and mast cells have all been shown to express granzyme B [77]. However, granzyme B expression in mast cells and basophils was not accompanied by perforin expression [77]. Granzyme B-secreting B cells possess cytotoxic potential in the absence of perforin expression [77]. Granzyme B expression in DCs may contribute to their cytotoxic potential via both perforin-dependent and perforin-independent pathways [77]. Granzyme A expression in human DCs depends on the developmental stage of DCs, since mature pDCs do not express granzyme A [77], and immature DCs eliminate CD8+ T cells via a perforin- and granzyme A-dependent mechanism [77].

Interestingly, in parvovirus B19-specific CD4+ T cells, a strong co-expression of granzyme B, perforin, and IL-17 [79] has been shown. Furthermore, by using an influenza-A model, Xie et al. [80] showed that all activated human T cells co-expressed IL-17 and granzyme B. Th17 cells are critically important for protective immunity of mucosal surfaces against fungal infections and extracellular bacteria but are also implicated in the pathology of several inflammatory and autoimmune diseases [81].

Apart from their HLA class II-restricted lytic activity, CD4+ CTLs can also release granzymes and perforin. Granzyme B has been shown to cleave auto-antigens and create unique fragments recognized by auto-antibodies [82]. Granzyme B can also function extracellularly [83] and mediates tissue destruction by degrading substrates, such as cartilage proteoglycans [84] and proteins involved in extracellular structure and function, such as vitronectin, fibronectin, and laminin [85].

Here, we demonstrate that CD4+ CTLs express cystatin F in both active monomeric and inactive dimeric form. Due to its glycosylation, dimeric form of cystatin F can be secreted and taken up by bystander cells and therefore regulate the activity of cysteine cathepsins in trans [33,39]. Increased extracellular cystatin F concentrations can impair the function of other effector cells, such as CD8+ CTLs and NK cells [33,42]. Although they were first considered an in vitro artefact, novel data suggest that CD4+ CTLs must be considered important effectors involved in autoimmune pathologies, controlling malignancies, and chronic infections [7,8,86]. Our novel data highlights the important role CD4+ CTLs may play in tumor-induced immune suppression.

## 4. Materials and Methods

### 4.1. Antibodies

For Western blotting, the following antibodies were used: rabbit anti-cystatin F (Sigma-Aldrich, St. Louis, MO, USA), rabbit anti-*N*-terminal part of cystatin F (Amsbio, Abingdon, UK), rabbit anti-β-actin (Sigma-Aldrich), rabbit and mouse anti-glyceraldehyde 3-phosphate dehydrogenase (GAPDH; Proteintech, Rosemont, IL, USA), rabbit anti-C/EBPα, mouse anti-cathepsin C, mouse anti-granzyme A, mouse anti-granzyme B, mouse anti-perforin, and mouse anti-legumain (B-8) (all from Santa Cruz Biotechnology, Dallas, TX, USA). The mouse anti-cathepsin H antibody 1D10 [87] and sheep anti-cathepsin L antibody [88] were prepared as described previously. The following secondary antibodies were used: anti-rabbit, anti-mouse, and anti-sheep secondary antibodies conjugated with horseradish peroxidase (HRP) (Jackson Immuno Research, West Grove, PA, USA); anti-rabbit and anti-mouse secondary antibodies conjugated with the fluorescent dyes DyLight 650 and DyLight 550, respectively (Invitrogen, Carlsbad, CA, USA); and anti-mouse secondary antibody conjugated with fluorescent dye StarBright 700 (Bio-Rad, Hercules, CA, USA).

### 4.2. Cell Cultures

Jurkat cells (CRL-11386; ATCC, Manassas, VA, USA) were cultured in Iscove’s Modified Dulbecco’s Medium (ATCC) with 10% fetal bovine serum (Gibco, Carlsband, CA, USA), 100 U/mL penicillin (Lonza, Basel, Switzerland), and 100 U/mL streptomycin (Lonza). The cells were treated with phorbol 12-myristate 13-acetate (PMA) together with the T-cell mitogen phytohemagglutinin (PHA) for 6, 24, and 48 h.

### 4.3. Cell Cloning

Cell cloning was performed by limiting dilution as previously described [43]. Clonal age is expressed in population doublings (PDs), estimated by cell counts at each subculture and counting the number of cumulative doublings. The culture medium used was supplemented with 100 U/mL IL-2 (purified natural product “Lymphocult T-HP”, gift of Biotest, Frankfurt). Cells were cryopreserved at different PDs, stored in liquid nitrogen and shipped in dry ice.

### 4.4. Preparation of Cell Lysates

For isolation of untouched naive CD4+ T helper cells (CD4+CD45RA+) and CD4+ memory T cells from PBMCs of healthy donors the Naive CD4+ T Cell Isolation Kit II, CD4+ Central Memory T Cell Isolation Kit, LS Columns, and a MiniMACS™ Separator were used (all from Miltenyi Biotech). Cells were washed in PBS, lysed in lysis buffer, incubated for 30 min on ice, and centrifuged at 16,000× *g* for 20 min at 4 °C. Supernatants were transferred to fresh tubes, and protein concentrations were determined using the DC-Protein Assay Kit (Bio-Rad). The lysis buffer for Western blot analysis comprised 50 mM Tris-HCl, pH 8, 150 mM NaCl, 1% Triton X-100, 0.5% sodium deoxycholate, 0.1% SDS, and 1 mM EDTA with added protease inhibitors (Roche, Basel, Switzerland). The lysis buffer for cathepsin activities comprised 0.1 M citrate buffer, pH 6.2, with 1% Triton X-100. The lysis buffer for granzyme activities comprised 25 mM HEPES, 250 mM NaCl, 2.5 mM EDTA, and 0.1% Nonidet p-40, pH 7.4.

Cell lysates (10 µg) from different T-cell clones were incubated with or without *N*-glycosidase F (PNGase F) or endoglycosidase H (EndoH) at 37 °C overnight according to the manufacturer’s instructions and were analyzed by non-reducing PAGE and cystatin F immunoblot.

### 4.5. Western Blotting

Samples containing 5–30 µg of cell lysate total protein were resolved with SDS-PAGE using 6% (Novex™ WedgeWell™ 6%, Tris-Glycine, Thermo Fisher Scientific), 12% TGX stain-free polyacrylamide gels (Bio-Rad) or 15% polyacrylamide gels and transferred onto nitrocellulose membranes using the trans-blot turbo transfer system (Bio-Rad) and trans-blot turbo RTA transfer kits with nitrocellulose membranes (Bio-Rad). TGX stain-free polyacrylamide gels were activated before the transfer with UV light for 1 min using the ChemiDoc MP System (Bio-Rad). After the transfer, the membranes were imaged for stain-free labelling of total proteins with the ChemiDoc MP System. Next, membranes were blocked in 5% non-fat dry milk in PBS (for cystatin F), 5% non-fat dry milk in tris-buffered saline with 0.1% Tween-20 (for Cat C, Cat H, Cat L, granzyme B and GAPDH), and 1.5% non-fat dry milk and 0.5% bovine serum albumin in tris-buffered saline with 0.1% Tween-20 (for perforin and granzyme A). The primary antibodies were diluted in blocking solution at 1:500 for cystatin F, 1:200 for Cat C, perforin, granzymes B and A, 1:600 for Cat H, and Cat L, and 1:1000 for GAPDH, and incubated overnight at 4 °C. After washing, the membranes were incubated with HRP- or fluorescently conjugated secondary antibodies diluted at 1:5000 in blocking solution. Membranes with HRP-conjugated antibodies were incubated with Lumi-Light Western blotting substrate (Roche). Images were acquired using a ChemiDoc MP System (Bio-Rad), and quantification analysis was performed in Image Lab, version 5.1 software (Bio-Rad). For quantification analysis, the experiments were repeated at least twice and GAPDH (cystatin F) or whole protein load (other proteins) were used for normalization.

### 4.6. Immunoprecipitation

Lymphocyte clones (10 × 10^6^) were washed with PBS and lysed in lysis buffer (50 mM Tris-HCl (pH = 7.4), 100 mM NaCl, 0.25% Triton X-100, and protease inhibitor cocktail (Roche, Germany)). After incubation on ice for 30 min, the cell lysates were centrifuged for 20 min at 16,000× *g*, and the supernatant was transferred to a new tube. Protein G Dynabeads (Invitrogen, USA) were coated with rabbit anti-cystatin F antibodies (3 µg, Davids Biotechnologie, Germany) and antibodies against lectin isolated from Macrolepiota procera (3 µg, BioGenes GmbH, Germany), which represented the negative control. Dynabeads complexed with antibodies were incubated overnight at 4 °C with 400 µL of cell lysates and 800 µL of lysis buffer. Dynabead immunoprecipitates were washed with ice-cold lysis buffer, resuspended in SDS loading buffer with 50 mM dithiothreitol, and denaturated by boiling at 100 °C for 10 min. The co-immunoprecipitated proteins were analyzed by Western blot.

### 4.7. Determination of Enzyme Activities

The enzyme activities were determined using acetyl-Ile-Glu-Pro-Asp-AMC (50 µM) and Z-Gly-Pro-Arg-AMC (200 µM; both from Bachem) for granzyme B and A, respectively. The following assay buffers were used: 50 mM Tris-HCl and 100 mM NaCl, pH 7.4 (for granzyme B) and 20 mM Tris and 150 mM NaCl, pH 8.1 (for granzyme A). Whole-cell lysates were first activated in assay buffer for 30 min at 37 °C. The substrate was then added, and the formation of fluorescent products was measured continuously on a microplate reader Infinite M1000 (Tecan, Männedorf, Switzerland).

## Figures and Tables

**Figure 1 ijms-22-08408-f001:**
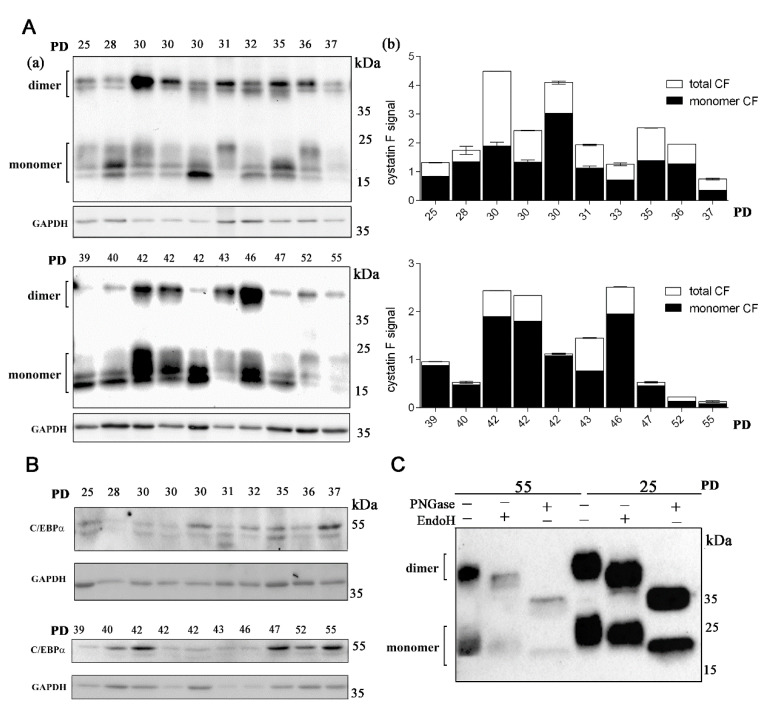
Cystatin F is expressed in long-term cultured CD4+ T cells. (**A**) Western blot analysis and cystatin F immunodetection in cell lysates of CD4+ T-cell clones (**a**). Representative Western blot out of two experiments and quantification for cystatin F signal. The data are represented mean and standard error of means (SEM). Quantification of the Western blot results shows total and monomeric cystatin F signal. GAPDH staining was used to show equivalent protein loading. Quantification was performed using Image Lab Software using GAPDH signal for normalization (**b**). (**B**) Western blot analysis and C/EBPα immunodetection in cell lysates of CD4+ T-cell clones. GAPDH staining was used to show equivalent protein loading. (**C**) Western blot analysis and cystatin F immunodetection in cell lysates of old (55) and young (25) T-cell clones after treatment with Peptide: *N*-glycosidase F (PNGase) and endoglycosidase H (EndoH).

**Figure 2 ijms-22-08408-f002:**
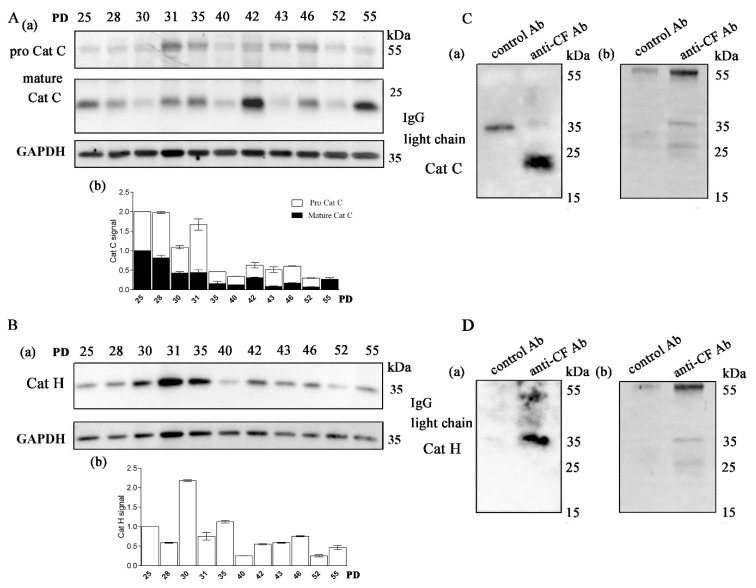
The main targets of cystatin F in long-term cultured CD4+ T cells are cathepsins C and H. (**A**) (**a**) Western blot analysis and cathepsin C (Cat C) immunodetection in cell lysates of CD4+ T-cell clones. (**b**) Quantification of the Western blot experiment shows mature Cat C and pro-Cat C signals. (**B**) (**a**) Western blot analysis and cathepsin H (Cat H) immunodetection in cell lysates of CD4+ T-cell clones. (**b**) Quantification of the Western blot experiment shows mature H signals. (**A**,**B**) Representative Western blot and quantification for cathepsins C and H. Quantification was performed using whole protein load for normalization and Image Lab Software. The data are represented as mean and standard error of means (SEM). (**C**) (**a**) Western blot analysis and Cat C immunodetection in immunoprecipitates with anti-cystatin F antibodies and negative control antibodies against lectin isolated from *Macrolepiota procera* for interaction of cystatin F with Cat C. (**b**) Imaging of stain-free activated protein membrane was used to confirm equal protein loading. (**D**) (**a**) Western blot analysis and Cat H immunodetection in immunoprecipitates with anti-cystatin F antibodies and negative control antibodies against lectin isolated from *Macrolepiota procera* for interaction of cystatin F with Cat H. (**b**) Imaging of stain-free activated protein membrane was used to confirm equal protein loading.

**Figure 3 ijms-22-08408-f003:**
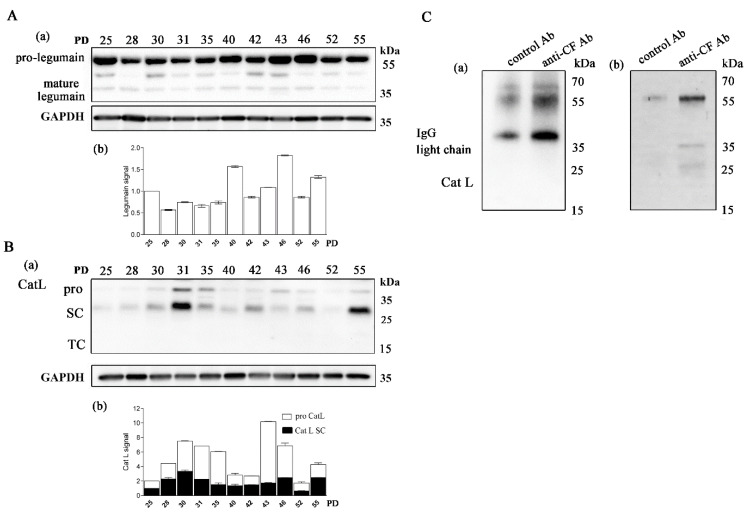
Long-term cultured CD4+ T cells express legumain and cathepsin L. (**A**) (**a**) Western blot analysis and legumain immunodetection in cell lysates of CD4+ T-cell clones. The pro-form (56 kDa) and mature form (36 kDa) of legumain are indicated. (**b**) Quantification of the Western blot experiment shows the legumain signal. (**B**) (**a**) Representative Western blot analysis and cathepsin L (Cat L) immunodetection in cell lysates of CD4+ T-cell clones. The mature Cat L pro-form (38 kDa), single-chain form (SC, 27 kDa) and the heavy chain of the two-chain form (TC, 20 kDa) are indicated. (**b**) Quantification of the Western blot experiment shows the mature Cat L pro-form and single-chain form. (**A**,**B**) Quantification was performed using whole protein load for normalization and Image Lab Software. The data are represented as mean and standard error of means (SEM). (**C**) (**a**,**b**) Western blot analysis and Cat L immunodetection in immunoprecipitates with anti-cystatin F antibodies and negative control antibodies against lectin isolated from *Macrolepiota procera* for interaction of cystatin F with Cat L. Imaging of stain-free activated protein membrane was used to confirm equal protein loading.

**Figure 4 ijms-22-08408-f004:**
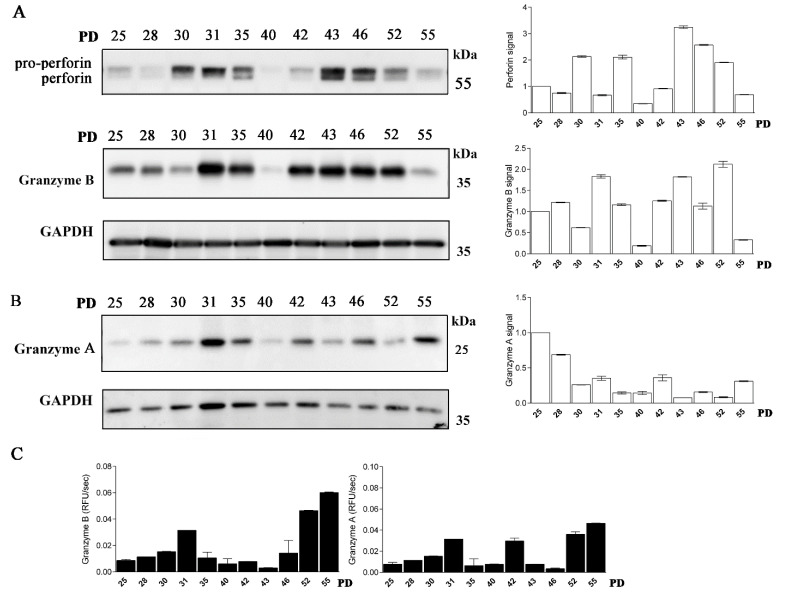
Long-term cultured CD4+ T cells express perforin and granzymes. (**A**) (**left**) Western blot analysis and perforin and granzyme B immunodetection in cell lysates of CD4+ T-cell clones. Pro-form of perforin and mature perforin are indicated. (**right**) Quantification of the Western blot experiment shows perforin and granzyme B signal. (**B**) (**left**) Western blot analysis and granzyme A immunodetection in cell lysates of CD4+ T-cell clones. (**right**) Quantification of the Western blot experiment shows granzyme A signal. Quantification was performed using whole protein load for normalization and Image Lab Software. (**B**) Representative Western blot analysis and granzyme A immunodetection in cell lysates of CD4+ T-cell clones. Quantification of the Western blot experiment shows granzyme A signal. (**A**,**B**) Quantification was performed using whole protein load for normalization and Image Lab Software. The data are represented as mean and standard error of means (SEM)). Western blots are representative of two repeated experiments. (**C**) Activities of granzyme B and granzyme A were determined in post-nuclear cell lysates. Values are mean and SD between triplicates.

## Data Availability

The whole western blot figures can be accessed in Appendix A.

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
