# Peer review of "Human CD4+ T-Cell Clone Expansion Leads to the Expression of the Cysteine Peptidase Inhibitor Cystatin F"

_ijms, 2021, doi:10.3390/ijms22168408_

Round 1

Reviewer 1 Report

The topic of the article is definitely interesting and it deals with potentially important hypothesis that aged CD4+ cells produce less cystatin F and thus the regulation of cathepsins that activate cytotoxic granzymes in lymphocytes is altered and the aged CD4+ cells becomes cytotoxic (CTL).

The authors employed Western blotting to show the variable presence of cystatin F, cathepsin C, H and L, granzymes A and B and perforin in CD4+ cells of different age (different number of population doubling). Morover, the immunoprecipitation was used to show that cystatin F binds to cathepsins.

Unfortunately I do not find the results very convincing. I am not able to see the pattern, where authors describe it. To me, the expression of all studied proteins varied erratically among cells with different PDs. The only way, how to prove the opposite, is proper statistical evaluation, which could be tricky and not trivial for such setting, but it is absolutely necessary. Without it, authors cannot use statements, such as “the content of cystatin F was generally higher in middle-aged lymphocyte clones…”, because it is not known, whether it is significant. And I would bet it is not. Also the ratio between monomeric and dimeric form of cystatin F and pro-enzymes and mature enzymes should be somehow evaluated. I am afraid whole setup must be redesigned in order to have comparable groups with sufficient N number, moreover the experiments should be reproduced at least in three independent experiments. It is not obvious to me, whether the triplicates are three independent experiments or biological replicates ((i.e. three wells in one plate or technical replicates (three wells in the gel). Some PDs have three lanes in the figure 1, some only one, nowhere is explained why it is like this.

Moreover I miss the comparison between regular CD4+ cells and CD4+ CTLs. If the question is, whether the expression of cystatin F and the activity of granzymes change during the aging, then also PDs lower than 25 should be shown as controls. If I understand correctly, the expression of cystatin F should be highest in regular CD4+ cells/ very young clones.

I think that the most important flaws of the study are insufficient number of independent experiments (or missing information that they were done) and missing statistical evaluation, which could show whether the conclusions are supported by statistically significant results. Unfortunately, the article should not be published in such state, because of flawed experimental design and missing statistics. Conclusions cannot be supported by the results.

Author Response

We are grateful to the reviewer for the opportunity to improve our manuscript. Please find our answers and additional figure in the attached document.

Reviewer 2 Report

The manuscript by Perišić Nanut et al, characterized the induced expression of cystatin F in combination with its target cathepsins C and H in long-cultured CD4+ T cells. Their data also shows that expression of cystatin F decreases in a manner dependent on cultured days while the activities of granzymes A and B increases. It is proposed that cystatin F is involved in regulating cytotoxicity of CD4+ T cells.

The aim of the study is clearly defined and the supporting results are presented in a well-organized way. Discussion on the relationship between each data could be improved by adding the experiment to examine if the manipulation of expressed level of cycstain F or any other enzymes could in fact regulate/modulate the cytotoxicity of the cells.

Minor comments:

Fig. 2C, 2D, and 3C:

Is it possible to add images of immunoprecipitated cystatin-F as a standardizing control.

Data are all about protein expression and enzymatic activities. Is it possible to show time dependent cytotoxicity change by other method? Is it possible to add a data examining the effect of secreted cystatin F?

Author Response

We are grateful to the reviewer for the opportunity to improve our manuscript. Please find our point by point answers below. Please also see the attached document where you will also find the additional Supplementary Figure 2. 

Comments and Suggestions for Authors

The manuscript by Perišić Nanut et al, characterized the induced expression of cystatin F in combination with its target cathepsins C and H in long-cultured CD4+ T cells. Their data also shows that expression of cystatin F decreases in a manner dependent on cultured days while the activities of granzymes A and B increases. It is proposed that cystatin F is involved in regulating cytotoxicity of CD4+ T cells.

The aim of the study is clearly defined and the supporting results are presented in a well-organized way. Discussion on the relationship between each data could be improved by adding the experiment to examine if the manipulation of expressed level of cycstain F or any other enzymes could in fact regulate/modulate the cytotoxicity of the cells.

Answer: The effects of secreted cystatin F on cytotoxicity of cytotoxic T cells and NK cells was shown in our recent publications by Prunk et al.2020, and Senjor et al, 2021, respectively. In the present paper we have shown that the glycosylation, which is crucial for cystatin F secretion and uptake (Colbert et al, 2009, Perišić Nanut et al, 2017) is similar to the glycosylation pattern of cystatin F in cytotoxic T cells and NK cells which were shown to secrete cystatin F (Senjor et al, in preparation).

Colbert JD, Plechanovová A, Watts C. Glycosylation directs targeting and activation of cystatin f from intracellular and extracellular sources. Traffic (Copenhagen, Denmark) 2009; 10:425-37.

Perišić Nanut M, Sabotič J, Švajger U, Jewett A, Kos J. Cystatin F Affects Natural Killer Cell Cytotoxicity. Frontiers in immunology 2017; 8:1459.

Prunk M, Perišić Nanut M, Jakoš T, Sabotič J, Švajger U, Kos J. Extracellular Cystatin F Is Internalised by Cytotoxic T Lymphocytes and Decreases Their Cytotoxicity. Cancers 2020; 12:3660.

Senjor E, Perišić Nanut M, Breznik B, Mitrović A, Mlakar J, Rotter A, et al. Cystatin F acts as a mediator of immune suppression in glioblastoma.  2021.

Minor comments:

Fig. 2C, 2D, and 3C:

Is it possible to add images of immunoprecipitated cystatin-F as a standardizing control.

Answer: An additional Immunoprecipitation and immunolabeling for cathepsin C was performed as well as the immunolabelling of cystatin F. In the Supplementary Figure 2, only the monomeric form of cystatin F can be seen due to the addition of DTT prior to preparation of the samples for SDS-PAGE. Multiple bands seen are a consequence of glycosylation of cystatin F.

Data are all about protein expression and enzymatic activities. Is it possible to show time dependent cytotoxicity change by other method? Is it possible to add a data examining the effect of secreted cystatin F?

Answer: In this paper we have focused on the expression of cystatin F as well as its target peptidases in CD4+ lymphocytes. We have shown for the first time that CD4+ T cells grown under conditions of repetitive stimulation start expressing cystatin F in addition to granzymes and perforin as well as the proteases involved in their processing and activation. We have shown that the glycosylation, which is crucial for cystatin F secretion and uptake (Colbert et al, 2009, Perišić Nanut et al, 2017) is similar to the glycosylation pattern of NK cells which were shown to secrete cystatin F (Senjor et al, in preparation). The effects of secreted cystatin F on cytotoxicity of NK cells were shown in our recent publication by Senjor et al, 2021.

Colbert JD, Plechanovová A, Watts C. Glycosylation directs targeting and activation of cystatin f from intracellular and extracellular sources. Traffic (Copenhagen, Denmark) 2009; 10:425-37.

Perišić Nanut M, Sabotič J, Švajger U, Jewett A, Kos J. Cystatin F Affects Natural Killer Cell Cytotoxicity. Frontiers in immunology 2017; 8:1459.

Senjor E, Perišić Nanut M, Breznik B, Mitrović A, Mlakar J, Rotter A, et al. Cystatin F acts as a mediator of immune suppression in glioblastoma.  2021.

Reviewer 3 Report

The submitted manuscript by Perisic Nanut et al., evaluates the fundamental cytotoxic machinery by human CD4+ T cells. 
Particularly, the authors focused on cystatin F, a protein inhibitor of cysteine cathepsins, synthesized by CD8+ CTLs and natural killer cells, thus affecting the cytotoxic efficacy of these cells.
However, there are some minor points that require clarification, as follows:
1) Results - Figure 1A
The authors showed the WB analysis of the cystatin F monomeric and dimeric forms at selected PD's. 
Noteworthy, at PD 30, they showed three different clones which exhibit a very different ratio, namely, in one clone monomeric cystatin F is the predominant form, whereas in another clone, the dimeric form. 
Do the authors have an explanation for this unusual behaviour or since PD 30 is a transition point between the "young" and "middle-aged", is more likely due to experimental variations among different experiments? The reported behaviour at PD 30 was consistently seen under those conditions, thus suggesting a random behaviour where any molecular form of cystatin F
could coexist? Regarding the other time points, since only one clone was shown for each PD (PDs< 30, PDs >31), do the authors found a consistent behaviour shown? 
Since these results on the differential expression of the molecular forms of cystatin F in human CD4+ T cells are very interesting it will be worth including this explanation in the Discussion section, as well.

2) Results - section 3.2 
The authors state that in CD4+ T-cell clones, cysteine cathepsins C and H were the major targets of cystatin F.
Do the authors know or tested other cysteine cathepsins with either endopeptidase or exopeptidase activity? What about the aspartic cathepsin D?

3) Results - section 3.3 
- The title is "Human primary..... cathepsin L and legumain". However, in Figure 3, panel A corresponds to legumain whereas panel B to cathepsin L. 
Therefore, in order to keep consistency among them, it is suggested to use the same order of appearance for the title (section 3.3) and Figure 3, respectively.

- In line 176 the authors state that "In "all" samples...the Cat L pro-form...". However, in Fig. 3B (Line 52) it is not detected at all, whereas it is almost undetectable in samples #25, 28, 48 and barely detected the SC or proforms in sample #40, etc. The authors should clarify the discrepancy. Otherwise, that statement holds true for "legumain" and not "Cat L", and if so, it should be corrected accordingly.  

4)  Results - section 3.4
Similarly as above (item 3), in order to keep consistency, it is suggested to use the same order of appearance for the title (section 3.4) and Figure 4 
(now granzyme - panel B and perforin - panel A), respectively.

5) Results - Figure 4A
- The WB analysis showing the endogenous expression of pro-perforin and perforin in human CD4+ T cells, is interesting (Figure 4A, main text lines 207-209). 
However, it is too speculative for any conclusion made only on that data (lines 209-210). Otherwise, in order to get reliable (quantitative) results, 
 antibodies that would recognize each one of the enzyme forms (zymogen and active forms, respectively) should be used. 

- as seen in Fig. 4A, the cut-off of PD's for the "middle-aged", would be more consistent with the whole data at PD 40 and not 43.

6) Results - Figure 4B
The data presented in the graph Granzyme A signal vs PD (right panel) does not reflect the signal's intensity shown by WB (left panel).
Please, confirm the data since if you state that the signal for PD 25 is 1.0 then the signal for PD 31 should be significantly higher and not below 0.5.

7) Discussion
In line 252, in order to avoid confusion, regarding the Cat L gene, since the correct term is CTSL1 and not CSTL, it is suggested to replace it, accordingly.

8) Supplementary Figures
Based on Supplementary Fig.1,  How do the authors support their statement (lines 288-289) if all positive controls are missing?

Author Response

We are grateful to the reviewer for the opportunity to improve our manuscript. Please find our point by point answers and the additional image in the attached document.

Reviewer 4 Report

major:

* clones are generated from healthy young adults. => what is the age span? is it possible to obtain the data along with age and sex of individual?

* should age and sex be regressed out in all statistical tests? motivate otherwise

* lines 97-99: no data reference. overall this paragraph feels like it comes out of nowhere. flow must be improved

* Figure 1Aa - number of samples missing. what does the error bars represent? see also figures

* please deposit all raw images to figshare, biostudies or similar, to enable others to validate the quantification. since the images are central to this study, I think their availability is crucial for FAIR

minor things:

* Figure 1B - says cysteine in image. should be CEBPa?

* which cleaves nearly all N-linked olig -- is this text colored?

* “pro-Cat C in vitro” - wrong font?

* Figures - editorial advice needed on the AB vs ab panel labelling. I find it tricky to follow at times

Author Response

We are grateful to the reviewer for the opportunity to improve our manuscript. Please find our answers below or in the attached document.

Comments and Suggestions for Authors

major:

* clones are generated from healthy young adults. => what is the age span? is it possible to obtain the data along with age and sex of individual?

Answer: The age span of the donors was from 26 years to 80, but the sex is not known for a few of the samples so although very interesting it is not possible to correlate these results to the sex of the individuals at this point. Considering the influence of the gender differences in the immune regulation it is a very interesting proposition for future study.

* should age and sex be regressed out in all statistical tests? motivate otherwise

Answer: It was previously established that the age and sex of the donors does not affect clone behaviour (Pawelec, G., Y. Barnett, E. Mariani, and R. Solana. "Human CD4+ T Cell Clone Longevity in Tissue Culture: Lack of Influence of Donor Age or Cell Origin." Exp Gerontol 37, no. 2-3 (2002): 265-9.). For example, the clone with PD 28 is from an, 80 y.o. woman, the clone with PD 25 is from a 26 y.o. woman and the clone with PD 55, is from and 45 y.o. woman. However, for the expression of cystatin F (Figure 1) there are differences between clones with the same PD and therefore we have included these observations in the Discussion.

* lines 97-99: no data reference. overall this paragraph feels like it comes out of nowhere. flow must be improved

Answer: For the statement “Furthermore, extracellular cystatin F, secreted by bystander cells, attenuates granzyme-mediated cytotoxicity in both NK and CD8+ T cells the references are provided. For the statement: “Here, we describe for the first time cystatin F expression in CD4+ T-cell clones together with the components of cytotoxic granules.” the reference is our new data presented here.

* Figure 1Aa - number of samples missing. what does the error bars represent? see also figures

Answer: The number of PD is indicated above on the western blot images. Quantification was performed in Image Lab Software. Error bars represent standard error of mean. This is now stated in the figure legends.

* please deposit all raw images to figshare, biostudies or similar, to enable others to validate the quantification. since the images are central to this study, I think their availability is crucial for FAIR

Answer: Quantification for most of the experiments was performed using whole Stain Free gels and whole protein load for normalization and Image Lab Software. The raw images have been deposited.

minor things:

* Figure 1B - says cysteine in image. should be CEBPa? Answer: Thank you - It should be CEBPa.

* which cleaves nearly all N-linked olig -- is this text colored? Answer: The text should not be colored, sorry

* “pro-Cat C in vitro” - wrong font? Answer: The font is now corrected.

* Figures - editorial advice needed on the AB vs ab panel labelling. I find it tricky to follow at times

Round 2

Reviewer 1 Report

I thank the authors for their responses and adjustments. Unfortunately, the design of the study, in my opinion, prevents from reliable results. Othewise, the study and experiments were performed solidly. Therefore, apart from my opinion on the study design, the manuscript can be published, if other reviewers do not see any problem.

Author Response

Due to technical limitations it was not possible to change the design of our study. However, we deeply believe that the original data presented in this paper will lead to a better fundamental understanding of cytotoxic CD4+T-cell biology and the role of cystatin F within the tumor microenvironment. We thank the reviewer for very useful suggestions and for helping us make the main message of the paper clearer. 

Reviewer 4 Report

My comments have been resolved, except I cannot find the link to the deposited raw data

Author Response

We are grateful to the reviewer for helping us improve our manuscript. The original files were sent directly to the Associate Editor and should be available for all the readers (please see the attached document).
